# Use of a mouse model for the isolation of *Borrelia puertoricensis* from soft ticks

Edwin Vázquez-Guerrero[ID]¹, Alexander R. Kneubehl², Guadalupe C. Reyes-Solís³, Carlos Machain-Williams⁴, Aparna Krishnavajhala², Paulina Estrada-de los Santos[ID]¹, Job E. Lopez²,⁵*, José Antonio Ibarra[ID]¹*

1 Departamento de Microbiología, Escuela Nacional de Ciencias Biológicas, Instituto Politécnico Nacional, Ciudad de México, México, 2 Department of Pediatrics, Baylor College of Medicine, Houston, Texas, United States of America, 3 Centro de Investigaciones Regionales Dr. Hideyo Noguchi, Universidad Autónoma de Yucatán, Mérida, México, 4 Estudios en Una Salud, Unidad Profesional Interdisciplinaria de Ingeniería Campus Palenque, Instituto Politécnico Nacional, Palenque, Chiapas, Mexico, 5 Department of Molecular Virology and Microbiology, Baylor College of Medicine, Houston, Texas, United States of America

* job.lopez@bcm.edu (JEL); jaig19@gmail.com, jibarrag@ipn.mx (JAI)

## Abstract

The isolation of tick-borne relapsing fever (TBRF) spirochetes has proven to be a useful tool to understand their distribution in geographic areas where the tick vectors inhabit. However, their isolation and culture are not easy and in general an animal model is needed to achieve this task. Here, argasid ticks were collected from a neighborhood in Ciudad Caucel, and they were identified as *Ornithodoros* (*Alectorobius*) *puertoricensis*. To determine whether these were infected with TBRF bacteria they were fed with healthy mice but only a low burden of spirochetes was observed. An immunosuppressed mouse model was used to feed the ticks suspected to be infected with spirochetes. After tick feeding, a higher number of bacteria was observed in blood samples, and spirochetes were successfully cultivated in Barbour-Stoenner-Kelly (BSK)-IIB media. Molecular analyses indicated that the isolate was *Borrelia puertoricensis,* while whole genome sequencing confirmed the finding. In summary, the present report shows that *A. puertoricensis* is present in Ciudad Caucel, an urban neighborhood in the outskirts of Merida city, and these ticks are infected with *B. puertoricensis*. Despite the fact that this species has not been directly associated with TBRF it represents a potential medical and veterinary health risk.

## Introduction

Tick-borne relapsing fever (TBRF) is caused by pathogenic spirochetes in the genus *Borrelia* and are primarily vectored in the Western Hemisphere by argasid ticks [1]. These vectors have a life cycle that complicates the detection of the *Borrelia* species they transmit [2]. Specially, the genus *Ornithodoros* feed within five to 60 minutes and subsequently return to the cavity in which they dwell, or under floor tiles and windowsills inside houses. In the American continent, TBRF is caused by *Borrelia hermsii, Borrelia parkeri, Borrelia turicatae, Borrelia mazzottii,* and *Borrelia venezuelensis*, which are all transmitted by different species of argasid ticks, while *Borrelia miyamotoi* is transmitted by ixodid ticks [3]. However, in Latin America, the distribution of argasids and the TBRF spirochete species they transmit has been unclear.

**Data availability statement:** All relevant data are in the manuscript.

**Funding:** This work was supported by funds to JAI from Secretaría de Investigación y Posgrado-IPN (20230850, 20240122, 20241496, 20250266) and by funds provided to JEL from the National School of Tropical Medicine at Baylor College of Medicine. The funders had no role in study design, data collection and analysis, decision to publish, or preparation of the manuscript.

**Competing interests:** The authors have declared that no competing interests exist.

While TBRF spirochetes were described in Mexico in the twentieth century, the disease is not recognized by healthcare providers [4]. Humans infected with TBRF spirochetes show nonspecific symptoms that include sporadic fever, headache, myalgias, chills, nausea, vomiting, rigors, and pregnancy complications [5]. TBRF is often misdiagnosed as Lyme disease, although no isolates of the latter pathogen have been described in Latin America.

There has been taxonomic confusion with argasid ticks from Latin America, causing a poor understanding of the public health impact of TBRF borreliosis. For example, in the early 1900s, a species of TBRF spirochete circulated in Panama causing significant morbidity [6]. *Ornithodoros talaje* was implicated as the vector [7], but recent records of this tick from Panama are absent. Another argasid from Panama that was not considered as the vector of TBRF spirochetes was *Alectorobius* (*Ornithodoros*) *puertoricensis,* which has previously been misidentified as *O. talaje* [8]. A comparative analysis of *O. talaje* and *A. puertoricensis* by Fox indicated that the two were synonymous [9]. Work by Venzal et al. further showed that adults and nymphs of *O. talaje* and *A. puertoricensis* are morphologically similar [10]. Recent collections of *A. puertoricensis* from Panama demonstrated that this species transmits *Borrelia puertoricensis* [2]. Collectively, these studies indicated that updated surveillance efforts are needed for *A. puertoricensis*.

In Mexico, *A. puertoricensis* has been described in the state of Veracruz [11], but it is unknown whether *B. puertoricensis* circulates in the country. We report surveillance efforts collecting ticks in the Ciudad Caucel neighborhood in Merida. Using an immunosuppressed mouse model, we describe the isolation of a *Borrelia* species from *A. puertoricensis* ticks. The genomic characterization of this spirochete identified it as *B. puertoricensis*. We also performed a comparative analysis of the diagnostic antigen *Borrelia* immunogenic protein A (BipA) across known species of TBRF spirochetes [12]. Our results demonstrated that *B. puertoricensis* extends to the southeastern part of Mexico and are found in areas with high human activity.

## Methods

### Sampling sites

In January 2023, argasid ticks were collected from a small park in the Ciudad Caucel neighborhood in Merida city (latitude 20.985954, longitude −89.695441), the capital of Yucatan (Fig 1A), hereafter referred as Ciudad Caucel. For the collection of the ticks, no permit was required as ticks are not endangered species in Mexico and the collection was done in a public park without disturbing any sections of this site. Opossum (*Didelphis virginiana*) were sighted in the neighborhood and four of their burrows were sampled using an aspirator or dry ice as a source of carbon dioxide to bait ticks, as previously described [13] (Fig 1B and 1C). Collected samples were placed in 50-ml ventilated centrifuge tubes and ticks from each burrow were kept separated. Permissions for collecting ticks was obtained from Secretaría de Medio Ambiente y Recursos Naturales from the Mexican government (permit number SGPA/ DGVS/03082/22). Permit for collecting in the park was not necessary as this is a public and open space.

### Argasid tick identification and classification

Live ticks were returned to our laboratory in Mexico City and maintained at 25 °C and 85% relative humidity inside a glass desiccator as described previously [13]. Ticks were speciated by using light microscopy and previously described taxonomical guides [14–16]. Molecular classification was done as follows: given that all collected adult ticks had a similar morphology genomic DNA was isolated from only a few of them (n = 3) with a DNeasy Blood and Tissue kit

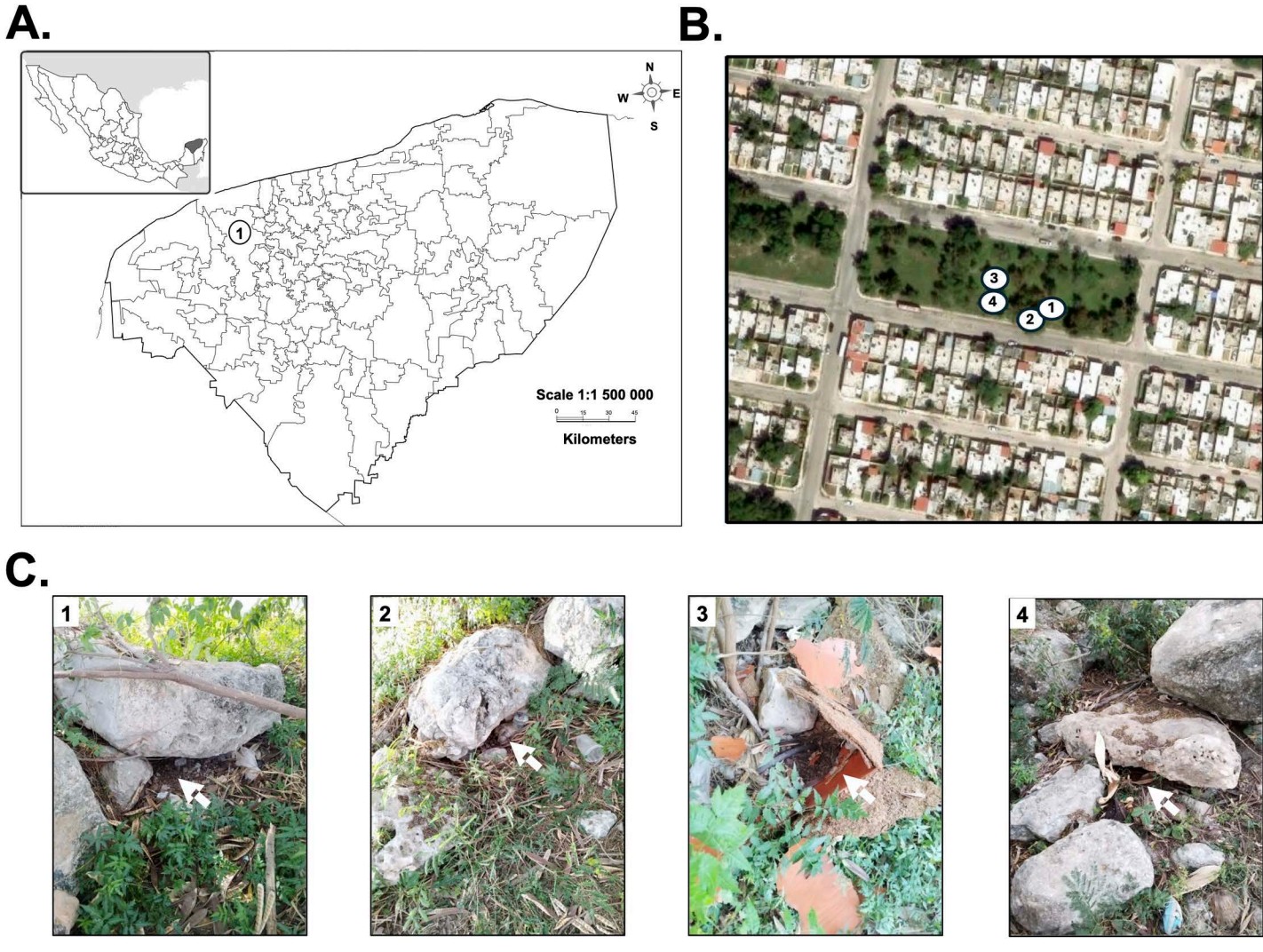

**Fig 1. Collection and identification of *Alectorobius* (*Ornithodoros*) *puertoricensis* in Ciudad Caucel.** Shown is a map of the state of Yucatan (A), the number 1 in panel A shows the location of Merida city (892,000 persons by the 2020 national census). The left upper corner map shows the location of this in Mexico (A). The collection sites were in Ciudad Caucel (B). The terrain map and the numbers indicate location of the burrows (B). The white arrows in (C), panels 1 through 4, point to where ticks were collected. In (C), C1 to C4 point to the respective locations in panel (B). Terrain maps were obtained from USGS EROS (Earth Resources Observatory and Science (EROS) Center) (public domain: http://eros.usgs.gov/#).

following the manufacturer's instructions (Qiagen, Hilden, Germany). A ~ 475 nucleotide region of the mitochondrial 16S rDNA gene was amplified by PCR as previously described [17,18] using the Tm 16S+1 (5'-CTGCTCAATGATTTTTTTAAATTGC-3') and Tm 16S-1 (5'-CCG GTCTGAACTCAGATCATGTA-3') primers. Amplification conditions were performed, as previously described [17,18]. PCR products were observed in 1% TAE agarose gels and further submitted for sequencing to obtain a 2X coverage at Macrogen Inc. (https://dna.macrogen. com/). Sequences were assembled and trimmed using MUSCLE v5 and ChromasPro (https:// technelysium.com.au/wp/chromaspro/) [19]. Consensus sequences obtained were compared with available sequences in GenBank using BLASTn (https://blast.ncbi.nlm.nih.gov). Nucleotide identity was assigned based on the expected value (e-value). A phylogenetic analysis using the maximum likelihood method was performed in PhyML 3.1 with the GTR + G model [20].

### Ethics statement

All animal procedures followed the guidelines of the Mexican Official Norms (NOM-062-ZOO-1999) for the care and use of laboratory animals. Protocols were approved by our Institutional Animal Care and Use Committee at Escuela Nacional de Ciencias Biológicas (Comité de Ética en Investigación, CEI-ENCB protocol # ZOO-001-2022e1).

### Animal model

To determine whether the collected ticks were infected by *Borrelia* spp. a mouse model was used as described previously with modifications [13]. Ticks were grouped in pools and fed on one of three different mouse strains (DBA/2J, C57BL/c or BALB/c) depending on the availability at the Instituto Politécnico Nacional. In total, 23 animals were used to feed from 3 to 24 ticks from each group (Table 1). Two and a half microliters of blood were collected daily from mouse tails for 18 consecutive days to evaluate the presence of spirochetes by dark field microscopy (Olympus CX33 Microscope), as described previously [2]. Spirochetes were counted per 6 µl of blood (dilution 1:2 with EDTA 1%) under dark-field illumination by using a Petroff–Hausser counting chamber (Hausser Scientific, Horsham, PA) with a depth of 0.02 mm [2]. When spirochetes were observed, the infected mouse was euthanized by administering sodium pentobarbital (150 mg/kg) followed by exsanguination with a heparinized syringe (syringe was filled with 1 ml heparin [1,000 U/ml] and then emptied, remaining heparin was enough to avoid blood coagulation) to quantify and isolate the bacteria as described [2]. If no bacteria were detected, the mice were kept alive, observed for 30 days, and then the animals were exsanguinated as described above. The blood samples were centrifuged at $5,400 \times g$ for two minutes to obtain the sera. Serum samples were used to evaluate antibody responses to *Borrelia* by immunoblotting, as previously described [21].

Using a chemically induced immunosuppression mouse model, we refed four cohorts of *A. puertoricensis* that were infected based on our findings from the first tick feedings [22,23]. Briefly, seven and four days prior to feeding ticks, 10 mg/kg dexamethasone diluted in sterile water was subcutaneously administered to one DBA/2J mouse and three BALB/c mice. Dexamethasone was also administered twice a week for the duration of the experiment and the presence of spirochetes was determined as described above. After seven days mice were sedated as described above and exsanguinated, blood was processed as described [2], and BSK-IIB media was inoculated with infected serum.

### Bacterial culture

When spirochetes were observed in murine blood, the animal was exsanguinated by intracardiac puncture with a heparinized syringe (see above) and whole blood was centrifuged at $500 \times g$ for 5 min. The plasma was removed and centrifuged again at $5,000 \times g$ for 10 min [24]. The pellet was resuspended in 1 ml of BSK-IIB media and used to inoculate 4 ml of medium [2], supplemented with 10 µg/ml rifampin, 4 µg/ml phosphomycin, and 0.5 µg/ml amphotericin B [25]. Cultures were incubated at 35ºC and 5% $CO_2$ for approximately eight days [26,27]. To assess bacterial growth, an aliquot of the culture was placed on a glass slide and observed by dark field microscopy. When 20 to 30 bacteria per microscopy field were detected and the media indicator changed to yellow, they were subcultured in fresh BSK-IIB media for storage and DNA isolation purposes.

### Bacterial identification and genomics

Spirochetes were grown for no more than two passages, and bacterial genomic DNA was isolated using a phenol-chloroform procedure. Pulsed field electrophoreses was performed to determine DNA quality, as described [28]. For an initial identification, a polymerase chain

**Table 1. Characteristics of collected ticks[a] and infected non-treated mice.**

| Den number [b] | Batch | Mouse strain | Stages[c] | Spirochetemia[d] |
|---|---|---|---|---|
| 1 | 1 | DBA/2J | 2♂, 8N | – |
| | 2 | DBA/2J | 8♂, 10N | – |
| | 3 | DBA/2J | 6♂, 11N | – |
| | 4 | DBA/2J | 4♂, 15N | – |
| | 5 | DBA/2J | 12♀ | – |
| | 6 | BALB/c | 4♂, 6N | – |
| 2 | 1 | DBA/2J | 3N | – |
| 3 | 1 | DBA/2J | 7♂, 6N | – |
| | 2[e,f] | DBA/2J | 16♂, 4N | $2.5 \times 10^4$ |
| | 3 | DBA/2J | 1♂, 7N | – |
| | 4 | DBA/2J | 20♀ | – |
| | 5 | BALB/c | 2♂, 10N | – |
| | 6 | BALB/c | 11♂, 8N | $2.5 \times 10^4$ |
| | 7 | BALB/c | 8♂,9N | – |
| | 8 | BALB/c | 11♂, 9N | – |
| | 9 | BALB/c | 8♂, 16N | $2.5 \times 10^4$ |
| | 10 | BALB/c | 17♀ | – |
| 4 | 1 | BALB/c | 20N | – |
| | 2 | BALB/c | 17N | – |
| | 3 | BALB/c | 15N | – |
| | 4 | BALB/c | 4♂, 16N | $2.5 \times 10^4$ |
| | 5 | C57BL/6 | 18N | – |
| | 6 | C57BL/6 | 2♀, 8N | – |
| Total 4 | 23 | 23 | 143A (51♀, 92♂), 216N[g] | 4 |

[a]Ticks were collected in Ciudad Caucel in January 2023.

[b]Number of the dens corresponds with those shown in Fig 1.

[c]A, adults; ♀, Females; ♂, Males; N, Nymphs; L, Larvae.

[d]Spirochetes per milliliter blood.

[e]Three specimens of this batch (2♀, 1N) were used for molecular analyzes and they were identified as *Ornithodoros puertoricensis*.

[f]*Borrelia puertoricensis* CAU1 isolate was obtained from this batch.

[g]These numbers represent the live ticks used in the study. Dead ticks were 107.

reaction (PCR) for the gene coding for the 16S rRNA and flagellin (*flaB*) was performed. PCR conditions consisted of initial denaturation at 95°C for 2 min followed by 35 cycles at 95°C for 30 sec, annealing at 55°C for 30 sec, and an extension at 72°C for 2 min; after the last cycle, a final extension was performed at 72°C for 5 min [18]. To confirm the expected molecular size for each amplicon, PCR products were observed in 1% TAE agarose gel electrophoresis and further submitted for sequencing to obtain a 2X coverage at Macrogen Inc. (https://dna.macrogen.com/). Sequences were assembled and trimmed using MUSCLE v5 [19] and ChromasPro (https://technelysium.com.au/wp/chromaspro/). Consensus sequences obtained were compared with available sequences in GenBank using BLASTn (https://blast.ncbi.nlm.nih.gov). Nucleotide identity was assigned based on the expected value (e-value). In cases of identical e-values, we assigned the gene to the sequence with the highest value. A phylogenetic analysis using the maximum likelihood method was performed in PhyML 3.1 with the GTR + G model [19]. Sequences were deposited in GenBank.

To determine plasmid content and DNA integrity 1 µg of purified genomic DNA was subjected to pulse field electrophoresis as previously described [18]. Long-read sequencing was performed using the Oxford Nanopore Technologies (ONT) Mk1B platform with the SQK-RBK114.96 library preparation kit and R10.4.1 flow cell. Short-read sequences were generated by SeqCenter (SeqCenter, Pittsburgh, Pennsylvania, USA) using an Illumina 2x150 library preparation kit. Plasmid-resolved genome assembly was generated by using short-reads to polish the long-read data, as previously reported [26] with modification. Due to the increase in accuracy for Oxford Nanopore Technologies' platform and to prevent over polishing of the assembly, which can introduce errors, we modified the polishing pipeline for genome assembly. The mean ONT coverage was 439x and the mean Illumina coverage was 236x. Using a previously established approach [26], completeness and QV scored (based on the Phred scale) were determined to be 99.89% and Q53.82, respectively. The assembly was annotated with NCBI's Prokaryotic Genome Annotation Pipeline.

## Results

### Capture and identification of *Alectorobius (Ornithodoros) puertoricensis*

Surveillance for argasid ticks was conducted in neighborhood parks in Ciudad Caucel, located in the at the outskirts of west side of the Merida city (Fig 1A). Opossum (*Didelphis virginiana*) burrows were in a small park in the center of the neighborhood (Fig 1B) and 466 soft ticks were collected from four burrows (Fig 1C, Table 1). The life stages included 279 nymphs, 28 larvae, and 159 adults (56 females and 103 males) (Table 1).

Morphological identification of adults and nymphs indicated that all collected ticks were *A. puertoricensis*. The morphology was consistent with the original descriptions of Fox (1947) [9], and with specimens previously reported in Mexico [16]. These included the absence of eyes, cheeks present, body oval, hood present and well developed, and tarsus I with subapical dorsal protuberance (Fig 2A, 2B).

Morphological speciation was complemented with molecular identification by analyzing 16S rDNA mitochondrial sequences. Total DNA from three ticks was obtained and analyzed (Ap1 to Ap3). BLASTn analysis indicated that sequences from the three samples were ~ 97%

**A.**

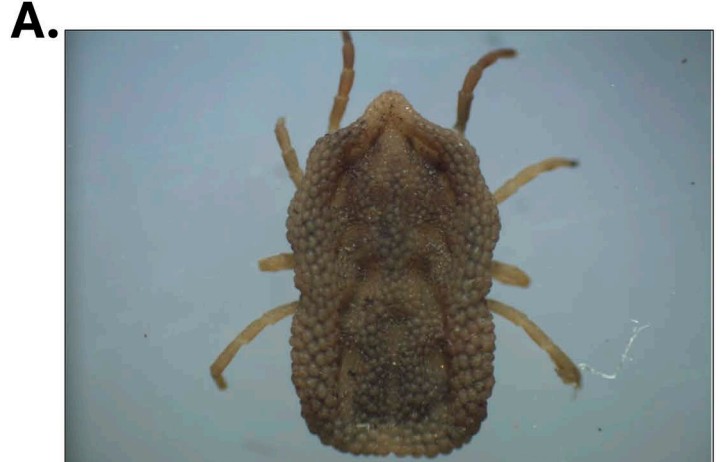

**B.**

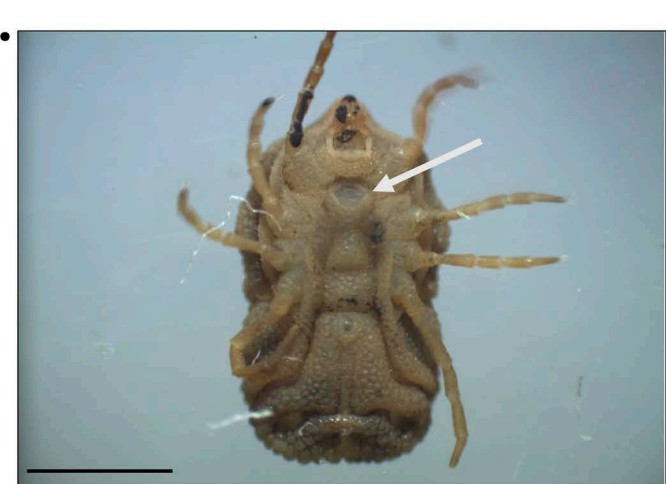

**Fig 2. Morphological and molecular identification of *A. puertoricensis*.** Stereo microscope images of a female *A. puertoricensis* specimen. The dorsal view is shown in (A) and the ventral in (B). The white arrow points to the genital aperture. Scale bars: 3 mm.

identical to *A. puertoricensis* from multiple regions of Central America and the Caribbean. The sequences were deposited in GenBank (OR710783-OR710785). A phylogenetic analysis of all three sequences corroborated that they grouped with *A. puertoricensis,* and they are in the same group as those reported from Mexico (Veracruz), Panama, Colombia, and Haiti (Fig 3). These results show that *A. puertoricensis* is present in Mexico in borrows located in urban parks and neighborhoods.

## Isolation of *Borrelia puertoricensis*

To determine whether the collected ticks were colonized by TBRF spirochetes, we fed them on mice. We observed a low level of spirochetemia ($2.5 \times 10^4$ bacteria per ml of blood) in four mice (Table 1) and failed to successfully culture the bacteria in Barbour-Stoenner-Kelly (BSK)-IIB medium. The four cohorts of ticks that transmitted spirochetes to mice were subsequently fed on clean mice under chemical immunosuppression to amplify the infection. We used one DBA/2J mouse and three BALB/c mice. In these mice we observed higher bacterial densities of $5 \times 10^5$ to $5 \times 10^6$ spirochetes per ml of blood. After seven days the four mice were exsanguinated, blood was processed as described [2], and BSK-IIB media was inoculated with infected serum. We successfully grew spirochetes that originated from the DBA/2J mouse in medium and generated glycerol stocks. This isolate was stored at −70 °C and designated Bp CAU1 (*B. puertoricensis* Ciudad Caucel isolate 1). DNA was extracted from the cultures and used for PCR amplification and sequencing of the *16S* rRNA coding gene and *flaB,* which indicated that the spirochetes were *B. puertoricensis* (Fig 4). Given these findings, we performed a whole genome analysis.

## Genomic organization of Bpu CAU1

Genomic DNA was isolated and spectrophotometry and pulsed field electrophoresis indicated that we obtained quality DNA for sequencing (521.6 ng/µl, 260/280 ratio of 2.16) [2,28]. The genome was assembled as reported with minor modifications [26]. The coverages were 439x and 236x, for the short-reads and long-reads, respectively. The assembled and annotated genome was deposited to NCBI´s GenBank (accession numbers CP149102-CP149123). A phylogenomic analysis of 650 single copy genes indicated that the Bp CAU1 isolate was *B. puertoricensis* (Fig 5A). The genome consists of a 928,012 bp linear chromosome, 16 linear plasmids (10,117 bp to 109,690 bp) and 5 circular plasmids (28,746 bp to 59,605 bp) (Fig 5B). Taken together, these results showed that a pool of the argasid ticks collected in Ciudad Caucel are colonized by *B. puertoricensis*.

## Identification and analysis of the diagnostic antigen, *Borrelia* immunogenic protein A (BipA)

Since evidence suggests that BipA is a species-specific antigen that can be used to determine vertebrate exposure to TBRF spirochetes [12], we assessed the CAU1 genome and identified a homolog. An *in silico* translation of *bipA* indicated that it had 97.8% and 71.6% amino acid identity with *B. puertoricensis* SUM (Panama) and *B. turicatae* CAM1 (Mexico), respectively (Table 2) [2,29]. We also observed interspecies diversity between TBRF spirochete isolates. The detection of a BipA homolog that is divergent from other known species (*B. turicatae*) in Mexico suggests that the antigen could be used in surveillance studies.

## Discussion

Despite that twenty-five species of *Ornithodoros* have been reported in Mexico [16], studies of TBRF spirochetes and their vectors have been neglected and forgotten. Three argasid

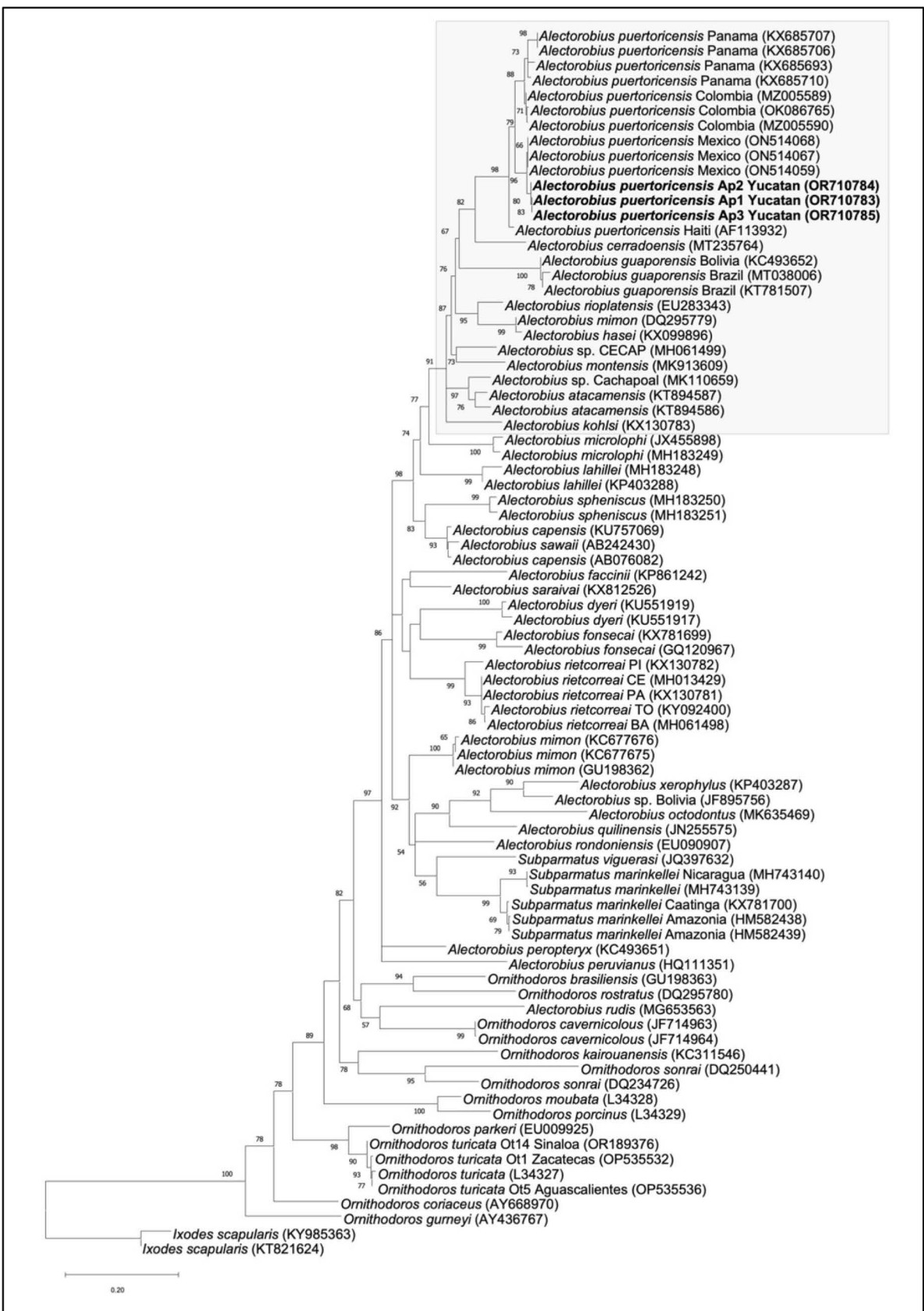

**Fig 3. Phylogenetic analysis of mitochondrial DNA sequence fragments from soft ticks *Alectorobius* spp.** The analysis was performed with the maximum likelihood method using the GTR + G model of nucleotide substitution. Bootstrap analysis, shown in branches, was

performed with 1000 replications. Shown in bold letters are the sequences of the analyzed ticks from this study. In the gray box is the group *talaje*. In parenthesis the accession numbers at NCBI. Bar displays the differences between sequences (0.20). *Ixodes scapularis* was used as an outgroup.

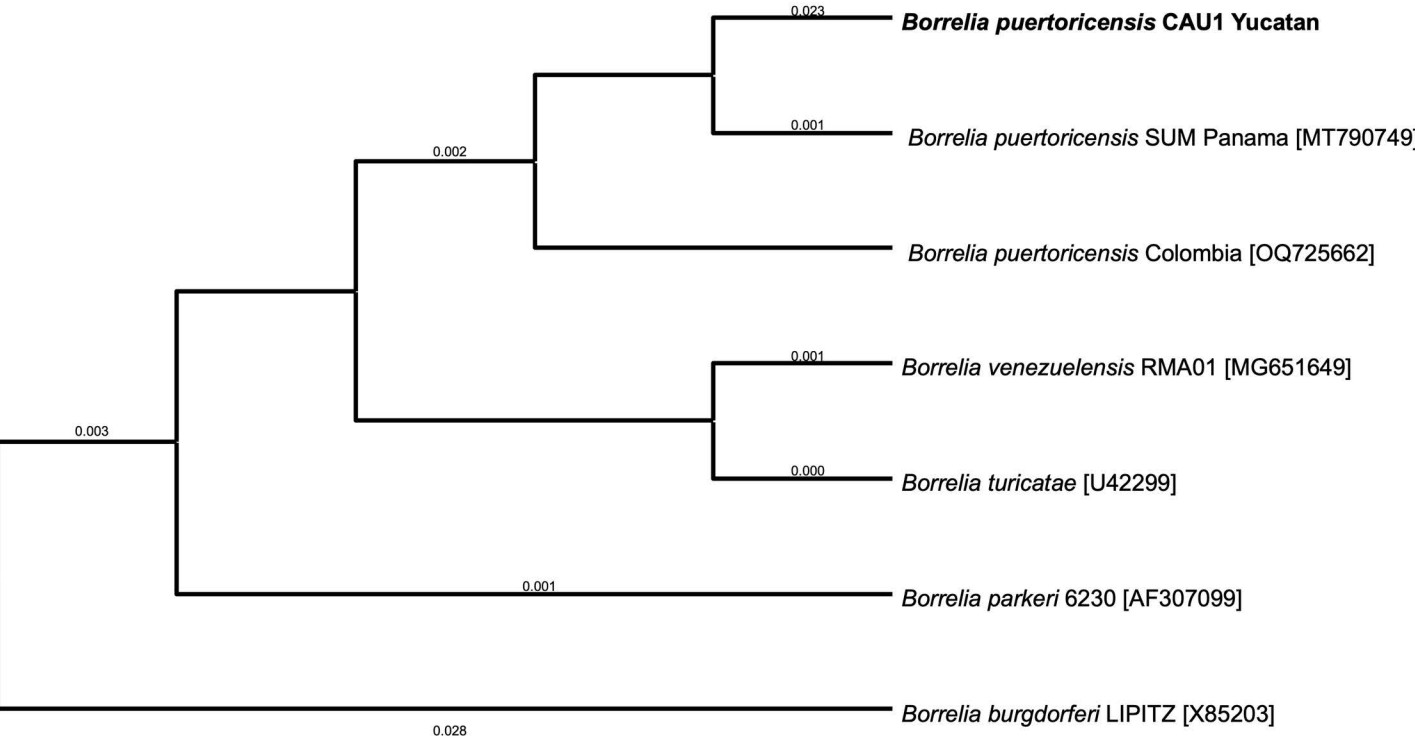

**Fig 4. Maximum-likelihood tree constructed from 16S rRNA partial sequences of various *Borrelia* species.** The analysis was performed with the amino acid substitution model Blosum62. Sequences generated in this study are in bold. Numbers represent bootstrap support generated from 1000 replications. The NCBI accession numbers are in brackets and *Borrelia burgdorferi* was used as outgroup.

species, *Ornithodoros turicata*, *Ornithodoros talaje*, and *Ornithodoros dugesi* are known vectors of TBRF spirochetes and transmit *Borrelia turicatae, B. mazzottii* and *B. dugesii,* respectively [3]. Although the presence of *A. puertoricensis* has already been described in Mexico [30], and more recently specimens had been collected in the Mexican state of Veracruz [11], the presence of *B. puertoricensis* has not been demonstrated. However, in Panama, this spirochete was successfully isolated using a MyD88(-/-) immunodeficient transgenic murine model [2].

Isolation of potential pathogens is important for confirmation of their circulation in an environment. For TBRF spirochetes, this is problematic for many reasons such as the vector ecology tick misidentification, lack of diagnostic tests and due to the difficulty culturing the bacteria. Even when an immunodeficient transgenic murine model has been stablish for the isolation of TBRF spirochete, this is not accessible in many laboratories around the World and thus *Borrelia* isolation continues to be a challenge. This is particularly true in laboratories where genetically immunosuppressed mouse strains are complicated to maintain, as in many Latin America and other resource-limited countries. Here we have standardized a model for the isolation of Borrelia sp. by chemically inducing immunodeficiency in mice.

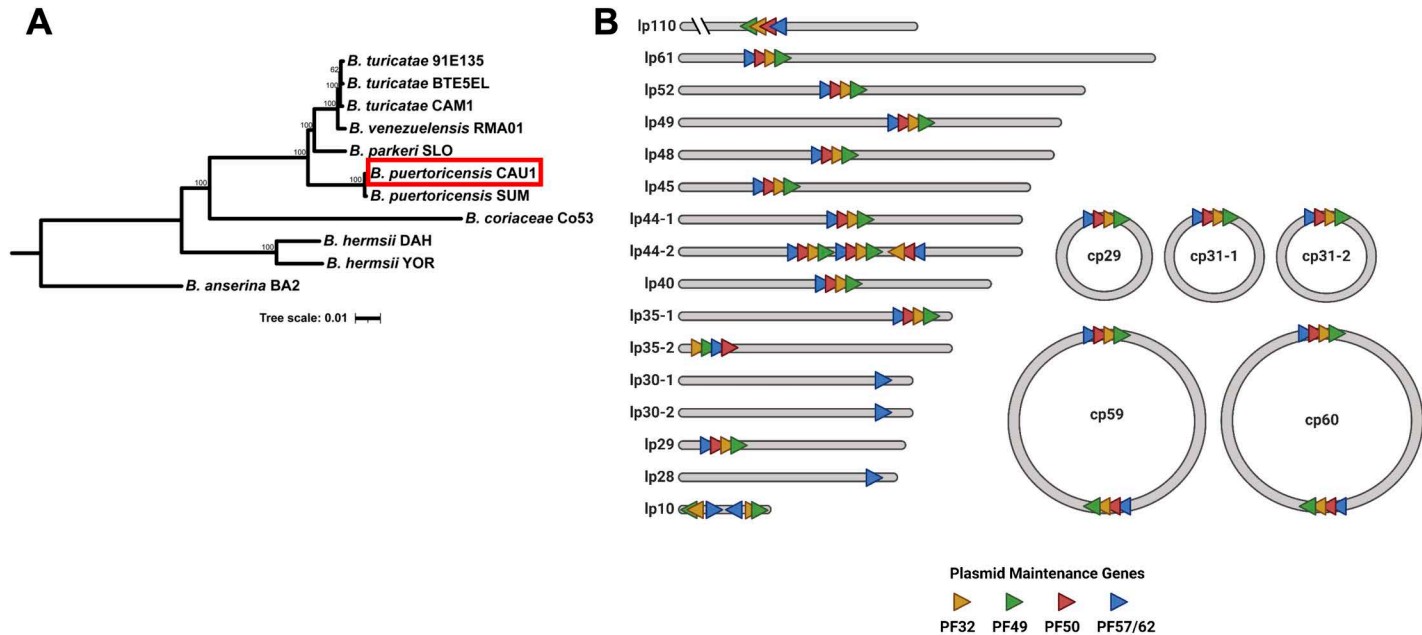

**Fig 5. Phylogenomic analysis of *B. puertoricensis* CAU1.** The isolate is shown boxed with a red rectangle and additional *Borrelia* genomes were included (A). The tree was generated with an edge-linked proportional partition model with 1,000 ultra-fast bootstraps. Scale bar indicates 0.02 substitutions per site. The genome of CAU1 is shown in (B) and is composed of 16 linear plasmids (lp) and 5 circular plasmids (cp). The numbers indicate the approximate size in base pairs and the colored triangles show the plasmid maintenance genes. PF, plasmid family.

**Table 2. Amino acid identity between BipA homologs[a,b].**

| | *B. rec* | *B. dut* | *B. cro* | *B. his* | *B. pue* | *B. pue* | *B. par* | *B. tur* | *B. tur* | *B. tur* | *B. ven* | *B. fai* | *B. nie* | *B. her* | *B. cor* | *B. miy* |
|---|---|---|---|---|---|---|---|---|---|---|---|---|---|---|---|---|
| *B. recurrentis* A1 | | 92.6 | 87.2 | 74.9 | 26.2 | 26.2 | 26.2 | 24.7 | 26.3 | 25.7 | 26.1 | 23.1 | 25.3 | 25.6 | 18.3 | 11 |
| *B. duttonii* CR2A | 92.6 | | 89.2 | 77 | 26.4 | 26.4 | 26.5 | 24.6 | 26.2 | 25.9 | 26.1 | 23.1 | 25.2 | 25.3 | 18.1 | 10.7 |
| *B. crocidurae* DOU | 87.2 | 89.2 | | 81.9 | 27 | 27 | 26.7 | 24.9 | 26.5 | 26.3 | 26.7 | 23.7 | 25.8 | 25.8 | 18.6 | 11.6 |
| *B. hispanica* CRI | 74.9 | 77 | 81.9 | | 26.1 | 26.1 | 26.4 | 25.2 | 26.6 | 26.3 | 26.7 | 22.8 | 24.9 | 25 | 18.4 | 12.1 |
| *B. puertoricensis* CAU1 | 26.2 | 26.4 | 27 | 26.1 | | 97.8 | 70.6 | 66.4 | 72 | 71.6 | 72.5 | 56.2 | 36 | 35.4 | 31 | 17.3 |
| *B. puertoricensis* SUM | 26.2 | 26.4 | 27 | 26.1 | 97.8 | | 70.1 | 66.4 | 72 | 71.4 | 72.5 | 56.2 | 35.8 | 35.9 | 31.5 | 18 |
| *B. parkeri* SLO | 26.2 | 26.5 | 26.7 | 26.4 | 70.6 | 70.1 | | 73.9 | 79.2 | 79 | 79.4 | 56.4 | 31.3 | 33.2 | 29.3 | 17.3 |
| *B. turicatae* BTE5EL | 24.7 | 24.6 | 24.9 | 25.2 | 66.4 | 66.4 | 73.9 | | 91.7 | 90.8 | 90 | 56.5 | 31.6 | 33.1 | 28.8 | 16.6 |
| *B. turicatae* 91E135 | 26.3 | 26.2 | 26.5 | 26.6 | 72 | 72 | 79.2 | 91.7 | | 97.7 | 96.8 | 60.5 | 34.2 | 35.7 | 31.3 | 18 |
| *B. turicatae* BTCAM1 | 25.7 | 25.9 | 26.3 | 26.3 | 71.6 | 71.4 | 79 | 90.8 | 97.7 | | 96.2 | 60.6 | 33.8 | 35.6 | 31.3 | 17.8 |
| *B. venezuelensis* RMA01 | 26.1 | 26.1 | 26.7 | 26.7 | 72.5 | 72.5 | 79.4 | 90 | 96.8 | 96.2 | | 60.9 | 34 | 35.3 | 31.9 | 18.1 |
| *B. fainii* Qtaro | 23.1 | 23.1 | 23.7 | 22.8 | 56.2 | 56.2 | 56.4 | 56.5 | 60.5 | 60.6 | 60.9 | | 32.2 | 30.6 | 27.5 | 16.9 |
| *B. nietonii* YOR | 25.3 | 25.2 | 25.8 | 24.9 | 36 | 35.8 | 31.3 | 31.6 | 34.2 | 33.8 | 34 | 32.2 | | 68.7 | 24.1 | 13.9 |
| *B. hermsii* DAH | 25.6 | 25.3 | 25.8 | 25 | 35.4 | 35.9 | 33.2 | 33.1 | 35.7 | 35.6 | 35.3 | 30.6 | 68.7 | | 25 | 16 |
| *B. coriaceae* Co53 | 18.3 | 18.1 | 18.6 | 18.4 | 31 | 31.5 | 29.3 | 28.8 | 31.3 | 31.3 | 31.9 | 27.5 | 24.1 | 25 | | 15.5 |
| *B. miyamotoi* CA17-2241 | 11 | 10.7 | 11.6 | 12.1 | 17.3 | 18 | 17.3 | 16.6 | 18 | 17.8 | 18.1 | 16.9 | 13.9 | 16 | 15.5 | |

[a]Numbers indicate the % of residues which are identical between the indicated proteins.

[b]TBRF *Borrelia* genera, species, and isolate are shown in the far-left column, and abbreviations were transposed into the top of each row. BipA amino acid sequences used to generate the table came from the following GenBank accession numbers: *B. recurrentis* A1 ACH95127.1, *B. duttonii* CR2A ETZ17992.1, *B. crocidurae* DOU AHH07353.1, *B. hispanica* CRI WP_235048026.1, *B. puertoricensis* CAU1 this study, *B. puertoricensis* SUM WP_247067590.1, *B. parkeri* SLO UPU87014.1, *B. turicatae* BTE5EL WP_119024356.1, *B. turicatae* 91E135 WP_192957046.1, *B. turicatae* BTCAM1 WP_330730264.1, *B. venezuelensis* RMA01 WP_247032594.1, *B. fainii* Qtaro WP_281862227.1, *B. nietonii* YOR WP_025434241.1, *B. hermsii* DAH WP_062705920.1, *B. coriaceae* Co53 WP_211228811.1, *B. miyamotoi* CA17-2241 ASQ29671.1.

Burgdorfer and Mavros showed that not all rodent species are competent hosts for TBRF spirochetes [31], and it was not surprising that immunosuppressed mice were needed for the isolation of *B. puertoricensis*. This *Borrelia* species has been challenging to propagate in Institute of Cancer Research (ICR), BALB/c, C57BL/6, and DBA/2J mouse strains [2]. In Panama, *A. puertoricensis* were fed on ICR mice and spirochetes were undetectable in the blood [2]. However, the mouse weakly seroconverted to TBRF *Borrelia* protein lysates, which suggested that the ticks were infected. The ticks were subsequently fed on a MyD88 -/- mouse, and spirochetes grew to high densities in this mouse strain. This enabled successful spirochete isolation in BSK-IIB medium. While MyD88 -/- mice were not available for our current study, chemically induced immunosuppression was a cost-effective approach.

The use of MyD88 -/- mice or chemically induced immunosuppression in wild type mice facilitates the propagation and amplification of TBRF spirochetes through different modes of action. Using a MyD88 -/- mouse model, Bolz and colleagues showed the dramatic propagation of *Borrelia hermsii* in the blood compared to wild type mice and a deficiency in clearing spirochetes [32]. Innate immunity is initiated when pathogen-associated molecular patterns like lipopolysaccharide, lipoproteins, unmethylated CpGDNA, and bacterial flagellin are bound to Toll-like receptors (TLRs), which activates the MyD88 cascade [33]. TLRs are highly expressed in dendritic cells, tissue macrophages, and mast cells, and activation of the MyD88 cascade results in pro-inflammatory cytokine production and initiation of adaptative immunity as well [33]. By using a mouse strain with impaired innate and humoral immunity, TBRF *Borrelia* reach high concentrations in blood, facilitating successful spirochete isolation [2,32].

Chemical immunosuppression is another approach to aid in the amplification of TBRF spirochete in murine blood. Newman and Johnson used cyclophosphamide, which suppresses B- and T-cells and prevented *Borrelia turicatae* clearance from mouse blood with spirochetes attaining ~10-fold higher densities compared to untreated mice [34]. Dexamethasone is a glucocorticoid that in rabbits generated a total decrease of lymphocytes in the blood [35]. Dexamethasone has also been used in mice for inducing immunosuppression during mycobacteria infection [23]. In our study, it is likely that this drug reduced both the innate immunity and the activation of B-cells in mice, allowing for the proliferation of *B. puertoricensis*. We are currently generating a colony of DBA/2J at the Escuela Nacional de Ciencias Biológicas that can be immune suppressed for isolation of *B. puertoricensis* from the remaining tick cohorts. Additionally, further work will define the lymphocyte populations and cytokines that are affected during dexamethasone treatment for amplifying TBRF *Borrelia*.

There is increasing evidence that *B. puertoricensis* is more prevalent in Latin America than previously thought [4]. This species was identified in argasid tick populations in Panama [2], and DNA from this bacterium was detected in opossum blood in Colombia [36]. Here we have shown that *B. puertoricensis* also circulates in the Yucatan peninsula of Mexico. Our work also demonstrated an extended range of the tick vector, *A. puertoricensis*. In addition to expanded tick collections, we recommend increasing surveillance efforts to determine vertebrate exposure to TBRF spirochetes. There have been advances using recombinant BipA (rBipA) as an antigen that can differentiate TBRF *Borrelia* species causing infection [12]. Increased surveillance efforts in Mexico will aid in defining the public health impact of *B. puertoricensis*.

## Acknowledgments

We deeply thank Dr. Lilia Dominguez-López and Dr. José Luis González-Quiroz from Escuela Nacional de Ciencias Biológicas-IPN for providing the mice for this study. We would also like to thank Dr. Wolfram Zueckert and Dr. Alan G. Barbour who were the original Editor and

reviewer, respectively, of this manuscript and were not reassigned when the manuscript was submitted for the second round of revision.

## Author contributions

**Conceptualization:** Edwin Vázquez-Guerrero, Job E. Lopez, J. Antonio Ibarra.

**Data curation:** Paulina Estrada-de los Santos, J. Antonio Ibarra.

**Formal analysis:** Edwin Vázquez-Guerrero, Alexander R. Kneubehl, Paulina Estrada-de los Santos, Job E. Lopez, J. Antonio Ibarra.

**Funding acquisition:** Job E. Lopez, J. Antonio Ibarra.

**Investigation:** Edwin Vázquez-Guerrero, Guadalupe C. Reyes-Solís, Carlos Machain-Williams, Aparna Krishnavajhala, Paulina Estrada-de los Santos, Job E. Lopez, J. Antonio Ibarra.

**Methodology:** Edwin Vázquez-Guerrero, Alexander R. Kneubehl, Guadalupe C. Reyes-Solís, Carlos Machain-Williams, Aparna Krishnavajhala, Paulina Estrada-de los Santos, Job E. Lopez.

**Project administration:** J. Antonio Ibarra.

**Resources:** Job E. Lopez, J. Antonio Ibarra.

**Software:** Alexander R. Kneubehl, Paulina Estrada-de los Santos.

**Supervision:** Paulina Estrada-de los Santos, Job E. Lopez, J. Antonio Ibarra.

**Validation:** Edwin Vázquez-Guerrero, Alexander R. Kneubehl.

**Visualization:** J. Antonio Ibarra.

**Writing – original draft:** Edwin Vázquez-Guerrero, Job E. Lopez, J. Antonio Ibarra.

**Writing – review & editing:** Edwin Vázquez-Guerrero, Alexander R. Kneubehl, Guadalupe C. Reyes-Solís, Carlos Machain-Williams, Aparna Krishnavajhala, Paulina Estrada-de los Santos, Job E. Lopez, J. Antonio Ibarra.

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
