## [Decision Letter · Decision Letter 0]

4 Jan 2025

PONE-D-24-47355Use of a mouse model for the isolation of Borrelia puertoricensis from ticks collected in Merida, Yucatan, MexicoPLOS ONE

Dear Dr. Ibarra,

Thank you for submitting your manuscript to PLOS ONE. After careful consideration, we feel that it has merit but does not fully meet PLOS ONE’s publication criteria as it currently stands. Therefore, we invite you to submit a revised version of the manuscript that addresses the points raised during the review process.

We look forward to receiving your revised manuscript.

Kind regards,

Maria Stefania Latrofa

Academic Editor

PLOS ONE

Journal Requirements:

This work was supported by funds to JAI from Secretaría de Investigación y Posgrado-IPN (20230850, 20240122, 20241496) and by funds provided to JEL from the National School of Tropical Medicine at Baylor College of Medicine. 

5. We note that your Data Availability Statement is currently as follows: All relevant data are within the manuscript and its Supporting Information files.

6. We note that Figure 1 in your submission contain [map/satellite] images which may be copyrighted. All PLOS content is published under the Creative Commons Attribution License (CC BY 4.0), which means that the manuscript, images, and Supporting Information files will be freely available online, and any third party is permitted to access, download, copy, distribute, and use these materials in any way, even commercially, with proper attribution. For these reasons, we cannot publish previously copyrighted maps or satellite images created using proprietary data, such as Google software (Google Maps, Street View, and Earth). For more information, see our copyright guidelines: http://journals.plos.org/plosone/s/licenses-and-copyright.

Additional Editor Comments:

Title: I would suggest removing "collected in Merida, Yucatan, Mexico " as the mouse model could be used to isolate ticks from other areas

Line 61: remove “This includes wild animal nests or dens”

Lines 135 and 192: remove “Sequences were deposited in GenBank.”, being reported below. The same goes for the “Annotation pipeline….(accession numbers CP149102-CP149123)”

Lines 248-249: Rewrite this sentence “These results show that A. puertoricensis is present in Mexico likely parasitizing opossums in urban neighborhoods”**, ** since as it stands it seems like a comment and not a result

Lines 329-330:  Change to “such as the vector ecology tick misidentification, lack of diagnostic tests and due to the difficulty culturing the bacteria”

Reviewers' comments:

Reviewer's Responses to Questions

**Comments to the Author**

1. Is the manuscript technically sound, and do the data support the conclusions?

Reviewer #1: Yes

2. Has the statistical analysis been performed appropriately and rigorously? 

Reviewer #1: N/A

3. Have the authors made all data underlying the findings in their manuscript fully available?

Reviewer #1: Yes

4. Is the manuscript presented in an intelligible fashion and written in standard English?

Reviewer #1: Yes

5. Review Comments to the Author

Reviewer #1: The present study investigates if Ornithodoros puertoricensis ticks might act as vectors for Borrelia puertoricensis, not being previously reported in Ciudad Caucel.

Line 122: Is there an explanation to analyse such a low number of ticks (n=3)? This number might be low to emphasize if a result is reliable and to further perform statistics in the future. Please explain and clarify.

Line 151: Please briefly describe how the exsanguination with a heparinized syringe to quantify and isolate the bacteria was performed

Line 155: Please briefly describe how the samples were analyzed to evaluate the antibody response against Borrelia

Line 156: Please specify what is a “chemically induced immunosuppression mouse model”, and how these animals were “prepared” for the experiments

Line 171: Please briefly justify why the choice of the temperature at 35ºC to cultivate the bacteria, once the ticks temperature may vary accordingly to the environment (being even higher/lower than 35ºC). Please specify if other temperatures were used to test the best culture conditions

Line 173: Please specify which was the expected concentration of bacteria considered “enough” for the subculturing and DNA isolation. Only saying that “when the bacteria were detected” may not be enough to the reader to repeat the procedures.

Line 274: I wonder whether if the extraction with the phenol-clorophorm might represent a good choice to perform genome sequencing. Where the information related to the quality of the analysis are showed? The authors are invited to clarify these points somewhere on the paper.

Line 331: Please specify which are the “Recent improvements in culture medium formulations” that have been made lately to clarify for the reader.

Line 332: Which may be the main reasons for the absence of which the accessible animals for the spirochete isolation? The authors are invited to describe these informations on the text.

Line 374: Please mention a reference that mention this information “There is increasing evidence that B. puertoricensis is more prevalent in Latin America”.

6. PLOS authors have the option to publish the peer review history of their article (what does this mean? ). If published, this will include your full peer review and any attached files.

**Do you want your identity to be public for this peer review?** For information about this choice, including consent withdrawal, please see our Privacy Policy .

Reviewer #1: No

---

## [Author Response · Author response to Decision Letter 0]

14 Jan 2025

Dear Dr. Latrofa

Thank you for your message and for taking over our manuscript. Before I do any more comments I would llike to tell you that this paper was originally edited by Dr. Wolfram Zueckert but the reviewed version couldn’t be uploaded because the Editorial Manager had a glitch that couldn’t be fixed and thus the PLOS ONE office asked me to send it as a new submission, when actually it wasn’t. Anyhow, I believe you needed to know this as thgis explains a bit my insistence.

Here I answer your comments, those made by the Journal and those by the reviewer, all are greatly appreciated for their efforts in this process. Please note that the reponses are marked as such and are placed immediately below the corresponding comment/suggestion.

Please do not hesitate to contact me in case there is any questions regarding this submission. I also take this opportunity to thank you and the reviewer for your comments.

Best regards,

Antonio

Additional Editor Comments:

Title: I would suggest removing "collected in Merida, Yucatan, Mexico " as the mouse model could be used to isolate ticks from other areas

RESPONSE: We agree with the Editor and modified the title but also added the that we isolated it from soft ticks, we hope she agrees with this change.

Line 61: remove “This includes wild animal nests or dens”

RESPONSE: We have removed this sentence as required.

Lines 135 and 192: remove “Sequences were deposited in GenBank.”, being reported below. The same goes for the “Annotation pipeline….(accession numbers CP149102-CP149123)”

RESPONSE: We have removed these sentences as required.

Lines 248-249: Rewrite this sentence “These results show that A. puertoricensis is present in Mexico likely parasitizing opossums in urban neighborhoods”, since as it stands it seems like a comment and not a result.

RESPONSE: We have modified this sentence by only mentioning that the ticks were located in urban parks and neighborhoods (line 252).

Lines 329-330: Change to “such as the vector ecology tick misidentification, lack of diagnostic tests and due to the difficulty culturing the bacteria”

RESPONSE: We have removed these sentences as suggested.

Reviewers' comments:

Reviewer's Responses to Questions

Comments to the Author

1. Is the manuscript technically sound, and do the data support the conclusions?

Reviewer #1: Yes

2. Has the statistical analysis been performed appropriately and rigorously?

Reviewer #1: N/A

3. Have the authors made all data underlying the findings in their manuscript fully available?

Reviewer #1: Yes

4. Is the manuscript presented in an intelligible fashion and written in standard English?

Reviewer #1: Yes

5. Review Comments to the Author

Reviewer #1: The present study investigates if Ornithodoros puertoricensis ticks might act as vectors for Borrelia puertoricensis, not being previously reported in Ciudad Caucel.

Line 122: Is there an explanation to analyse such a low number of ticks (n=3)? This number might be low to emphasize if a result is reliable and to further perform statistics in the future. Please explain and clarify.

RESPONSE: Thank you for this comment/question. Only three specimens were used to extract DNA because the rest were used in the mouse model for the isolation of Borrelia sp. Moreover, they were all collected from the same park and morphologically they are similar same. We are confident that the results obtained with these specimens are representative for the other adult ticks. We have added a statement about this in lines 123 to 124.

Line 151: Please briefly describe how the exsanguination with a heparinized syringe to quantify and isolate the bacteria was performed

RESPONSE: We have added this information in the results section (lines 155 to 157).

Line 155: Please briefly describe how the samples were analyzed to evaluate the antibody response against Borrelia

RESPONSE: The reference quoted in this section has all the protocols. Given that we didn’t present results on this we decided not to include the detailed protocol and only refer it. We hope that the reviewer agrees with us.

Line 156: Please specify what is a “chemically induced immunosuppression mouse model”, and how these animals were “prepared” for the experiments

Response: The immunosupression of the mice is mentioned in lines 164 to 168 in the same section.

Line 171: Please briefly justify why the choice of the temperature at 35ºC to cultivate the bacteria, once the ticks temperature may vary accordingly to the environment (being even higher/lower than 35ºC). Please specify if other temperatures were used to test the best culture conditions.

Response: Borrelia sp. growth has been described in related papers by our labs and others to ocurr at 34 or 35 oC. In our labs we decided to use 35oC. We have added more references to support this. No other conditions were tested as Dr. Lopez lab has this protocol well standarized and in our lab in Mexico we reproduced it.

Line 173: Please specify which was the expected concentration of bacteria considered “enough” for the subculturing and DNA isolation. Only saying that “when the bacteria were detected” may not be enough to the reader to repeat the procedures.

Response: We apologize for this. We have added information to clarify this in lines 180 to 181.

Line 274: I wonder whether if the extraction with the phenol-clorophorm might represent a good choice to perform genome sequencing. Where the information related to the quality of the analysis are showed? The authors are invited to clarify these points somewhere on the paper.

Response: In our laboratories experiences (Job Lopez’s and Antonio Ibarra’s labs) the DNA is of great quality to continue for sequencing with the two platforms (Nanopore and Illumina) mentioned in the manuscript. We have added this information in lines 290-291.

Line 331: Please specify which are the “Recent improvements in culture medium formulations” that have been made lately to clarify for the reader.

RESPONSE: We have decided to remove this statement as the possibility of culturing Borrelia is not that recent. We apologize for this.

Line 332: Which may be the main reasons for the absence of which the accessible animals for the spirochete isolation? The authors are invited to describe these informations on the text.

RESPONSE: We have a modified this section and added more text that complements the following sentence (lines 336 to 340) together with information that was already in the manuscript (lines 340-344).

Line 374: Please mention a reference that mention this information “There is increasing evidence that B. puertoricensis is more prevalent in Latin America”.

RESPONSE: We have included a review reference that mentions the distribution of Borrelia puertoricensis in this region (line 382).

6. PLOS authors have the option to publish the peer review history of their article (what does this mean?). If published, this will include your full peer review and any attached files.

Do you want your identity to be public for this peer review? For information about this choice, including consent withdrawal, please see our Privacy Policy.

Reviewer #1: No

---

## [Editor Report · Decision Letter 1]

21 Jan 2025

Use of a mouse model for the isolation of Borrelia puertoricensis from soft ticks

PONE-D-24-47355R1

Dear Dr. Ibarra,

We’re pleased to inform you that your manuscript has been judged scientifically suitable for publication and will be formally accepted for publication once it meets all outstanding technical requirements.

Kind regards,

Maria Stefania Latrofa

Academic Editor

PLOS ONE

---

## [Editor Report · Acceptance letter]

PONE-D-24-47355R1

PLOS ONE

Dear Dr. Ibarra,

I'm pleased to inform you that your manuscript has been deemed suitable for publication in PLOS ONE. Congratulations! Your manuscript is now being handed over to our production team.

Kind regards,

on behalf of

Dr. Maria Stefania Latrofa

Academic Editor

PLOS ONE